Plants from the abandoned Nacozari mine tailings: evaluation of their phytostabilization potential

Santos Alina E. 1
Cruz-Ortega Rocio 2
Meza-Figueroa Diana 3
Romero Francisco M. 4
Sanchez-Escalante Jose Jesus 5
Maier Raina M. 6
Neilson Julia W. 6
Alcaraz Luis David 7
Molina Freaner Francisco E. freaner@unam.mx 1 8
1 Departamento de Ecologia de la Biodiversidad, Instituto de Ecologia, Universidad Nacional Autonoma de Mexico , Hermosillo , Sonora , Mexico
2 Departamento de Ecologia Funcional, Instituto de Ecologia, Universidad Nacional Autonoma de Mexico , Ciudad de Mexico , Mexico
3 Departamento de Geologia, Universidad de Sonora , Hermosillo , Sonora , Mexico
4 Departamento de Geoquimica, Instituto de Geologia, Universidad Nacional Autonoma de Mexico , Ciudad de Mexico , Mexico
5 Herbario USON, Departamento de Investigaciones Cientificas y Tecnologicas, Universidad de Sonora , Hermosillo , Sonora , Mexico
6 Department of Soil, Water and Environmental Science, University of Arizona , Tucson , AZ , United States of America
7 Laboratorio Nacional de Ciencias de la Sostenibilidad, Instituto de Ecologia, Universidad Nacional Autonoma de Mexico , Ciudad de Mexico , Mexico
8 Estacion Regional del Noroeste, Instituto de Geologia, Universidad Nacional Autonoma de Mexico , Hermosillo , Sonora , Mexico
Davies Gwilym
Electronic publication date: 2017 May 4
Publication date: 2017
Volume: 5
Electronic Location ID: e3280
Received 2016 Aug 6; Accepted 2017 Apr 6
Copyright: ©2017 Santos et al.
Copyright year: 2017
Copyright holder: Santos et al.
License: This is an open access article distributed under the terms of the Creative Commons Attribution License, which permits unrestricted use, distribution, reproduction and adaptation in any medium and for any purpose provided that it is properly attributed. For attribution, the original author(s), title, publication source (PeerJ) and either DOI or URL of the article must be cited.
License URL: https://creativecommons.org/licenses/by/4.0/

Keywords: Phytostabilization, Copper mine tailings, Sonora, Mexico

Funding: University of Arizona-Universidad Nacional Autonoma de Mexico Consortium on Drylands Research Programa de Apoyo a Proyectos de Investigación e Innovacion Tecnologica UNAM-PAPIIT-IN209015 National Institute of Environmental Health Sciences Superfund Research Program 2 P42 ES04940 This research was supported by the University of Arizona-Universidad Nacional Autonoma de Mexico Consortium on Drylands Research, the Programa de Apoyo a Proyectos de Investigación e Innovacion Tecnologica (UNAM-PAPIIT-IN209015), and the National Institute of Environmental Health Sciences Superfund Research Program Grant 2 P42 ES04940. The funders had no role in study design, data collection and analysis, decision to publish, or preparation of the manuscript.

==============================
Phytostabilization is a remediation technology that uses plants for in-situ stabilization of contamination in soils and mine tailings. The objective of this study was to identify native plant species with potential for phytostabilization of the abandoned mine tailings in Nacozari, Sonora in northern Mexico. A flora of 42 species in 16 families of angiosperms was recorded on the tailings site and the abundance of the most common perennial species was estimated. Four of the five abundant perennial species showed evidence of regeneration: the ability to reproduce and establish new seedlings. A comparison of selected physicochemical properties of the tailings in vegetated patches with adjacent barren areas suggests that pH, electrical conductivity, texture, and concentration of potentially toxic elements do not limit plant distribution. For the most abundant species, the accumulation factor for most metals was <1, with the exception of Zn in two species. A short-term experiment on adaptation revealed limited evidence for the formation of local ecotypes in Prosopis velutina and Amaranthus watsonii. Overall, the results of this study indicate that five native plant species might have potential for phytostabilization of the Nacozari tailings and that seed could be collected locally to revegetate the site. More broadly, this study provides a methodology that can be used to identify native plants and evaluate their phytostabilization potential for similar mine tailings.

Introduction

Unreclaimed mine tailings represent an important environmental problem as aeolian dispersion and water erosion may transfer potentially toxic elements into local trophic webs and nearby human settlements (Mendez & Maier, 2008). Phytostabilization is a form of remediation that involves the use of plants for in-situ stabilization of tailings and contaminants (Mendez & Maier, 2008). Implementing phytostabilization in a particular tailing requires identification of suitable plant species for specific ecological conditions as well as the appropriate amendments to allow plant germination and growth. In arid and semiarid environments, plants suitable for phytostabilization should be native, drought-, salt- and metal-tolerant and should limit shoot metal accumulation (Mendez & Maier, 2008). However, in highly modified novel ecosystems (sensu Hobbs, Higgs & Harris, 2009), non-native species could be considered if they are not invasive and provide desirable levels of phytostabilization.

Surveys of native plants that naturally colonize mine tailings offer the opportunity to identify species with potential in phytostabilization (Carrillo-Gonzalez & Gonzalez-Chavez, 2006; Cortes-Jimenez et al., 2013). Using native species has several advantages including adaptation to local environmental conditions and avoiding the introduction of invasive species that may affect local plant communities. Analysis of the pattern of metal accumulation among plants growing spontaneously in mine tailings allows identification of species with the best potential for phytostabilization (Santos-Jallath et al., 2012; Cortes-Jimenez et al., 2013). If bioconcentration (total element concentration in shoot tissue ÷ total element concentration in mine tailings) or accumulation factors are greater than 1, species are not suitable for phytostabilization but may have potential for phytoextraction; in contrast, if these ratios are less than one and if metal concentrations in plant tissues do not reach animal toxicity levels, species have potential for phytostabilization (Mendez & Maier, 2008).

Studies on the relationships between plant abundance and the physicochemical properties of mine waste have identified some of the factors that limit plant establishment in mine tailings (Conesa, Faz & Arnaldos, 2006; Anawar et al., 2013; Parraga-Aguado et al., 2013). Some of the tailing properties that have been found to influence plant abundance and distribution include pH (Conesa, Faz & Arnaldos, 2006), salinity (Parraga-Aguado et al., 2013) and metal concentration (Ortiz-Calderon, Alcaide & Li-Kao, 2008) and bioavailability (Kidd et al., 2007; Perlatti et al., 2015). Identifying the physicochemical properties of tailings that restrict plant establishment may be critical for the implementation of phytostabilization and for understanding the type of amendments needed (i.e., compost-assisted) to facilitate seedling establishment (Parraga-Aguado et al., 2013; Gil-Loaiza et al., 2016).

Studies of plant adaptation to local mine wastes where metals are bioavailable have important implications for the implementation of phytostabilization. The mechanism involved in the evolution of metal tolerance has been documented in many plant species (Antonovics, Bradshaw & Turner, 1971; Baker, 1987). Metal tolerant populations evolve through natural selection in response to high levels of metals in mine wastes (Ke et al., 2007). If metal tolerant ecotypes have evolved locally, phytostabilization should take advantage of such local ecotypes as they provide better cover and persistence than commercial varieties when grown on mine wastes (Smith & Bradshaw, 1979). Thus, the implementation of phytostabilization should take into account whether local plant ecotypes have evolved in particular mine tailings.

Past and current mining activities in northern Mexico have generated large amounts of unconfined mine wastes that poses risks to human populations and adjacent ecosystems (Jimenez, Huante & Rincon, 2006). The Nacozari region in northeastern Sonora hosts important copper deposits and one of the most important copper mines in northwestern Mexico. The Moctezuma Copper Company operated the Pilares mine east of Nacozari from 1900 to 1949 and generated several million tons of waste distributed in three tailings deposits that cover 52 ha around the town (Alvarado & Volke, 2004; De la O-Villanueva et al., 2013). Selected physicochemical parameters from these tailings have been described, including mineral composition (Romero et al., 2008), texture (De la O-Villanueva et al., 2013), pH, electrical conductivity and metal content (Meza-Figueroa et al., 2009). Although metal concentration in tailings is relatively low, the seasonal formation of efflorescent salts represents a serious problem; these salts can have high metal concentrations and are subject to wind dispersion which can move toxic metals from the tailings into nearby residential soils (Meza-Figueroa et al., 2009). Meza-Figueroa et al. (2009) suggest that one alternative for prevention of off-site aeolian dispersion and water erosion of the tailings is to create a vegetative cap using native plants. However, knowledge of the native plants growing near or on the Nacozari tailings and their potential in phytostabilization is limited.

Native plants have colonized the Nacozari tailings, although the distribution is localized and patchy and is not sufficient to prevent aeolian dispersion or water erosion. The overall goal of this study is to identify plant species with potential for phytostabilization of this site. Our objectives are to: (a) describe the taxonomic composition of the plant species growing in the Nacozari tailings; (b) describe the abundance and population structure of the most common species; (c) explore the physicochemical parameters that may limit plant distribution; (d) describe the pattern of metal accumulation in the most common plant species and (e) evaluate through short-term experiments whether local ecotypes have evolved in two of the plant species.

Material and Methods

Study area

The Nacozari mining district is located in northeastern Sonora, 123 km south of the Arizona (USA) border. The regional climate is semi-arid (BS1) with the mean temperature in Nacozari ranging from 12.1 °C during January to 27.9 °C during June. Mean annual rainfall is 578 mm with more than 60% occurring during July, August and September (Servicio Meteorologico Nacional, 2015). Regional vegetation is foothills thornscrub at lower elevations and oak woodland at higher elevations (Martinez-Yrizar, Felger & Burquez, 2010). Around Nacozari, common species from the thornscrub vegetation include Acacia constricta, Acacia farnesiana, Fouquieria splendens, Stenocereus thurberi and Mimosa dysocarpa. Along the Nacozari River, common species of the riparian vegetation include Prosopis velutina, Parkinsonia aculeata, Baccharis sarothroides, Baccharis salicifolia and Acacia farnesiana. According to the Nacozari soil chart H12-6 (INEGI, 2006), local soils include Leptosols, Phaeozems and Regosols. Meza-Figueroa et al. (2009) described some properties of the background soils north of Nacozari. Cu contents from background soils ranged from 48 to 96 mg/kg, whereas residential soils contained higher Cu-values of 200  mg/kg due to proximity of tailings.

The mining district hosts important ore deposits, including porphyry copper, breccia pipe and veins with Cu, Mo, Au, Ag and Zn (Alvarado & Volke, 2004). The Pilares copper ore deposit (0.7–1.2% Cu) was discovered in 1886 and purchased years later by the Moctezuma Copper Company (Phelps Dodge subsidiary). Mining activity lasted around 50 years, producing around 3,000 ton/day of copper until the mine was closed in 1949. Large amounts of waste were distributed into three deposits around the town of Nacozari and then abandoned (Alvarado & Volke, 2004).

The three tailing deposits differ in size: (1) the central tailings deposit is medium-sized and located within the urban area, at the southern margin of town, (2) the southern deposit is the smallest and is located just south of Nacozari along the Moctezuma-Agua Prieta road, and (3) the southeastern deposit is the largest and is located along the Nacozari-La Caridad road (Alvarado & Volke, 2004). Although the three deposits have been studied (Alvarado & Volke, 2004), work has concentrated on the central tailings deposit because of its proximity to the town (Meza-Figueroa et al., 2009). The mineral composition of this deposit is mainly quartz (SiO2), gypsum (CaSO4H2O), lepidocrocite (FeO[OH]) and copper sulfate (CuSO4, Romero et al., 2008). Mean pH is 3.8 ± 0.3 and mean electrical conductivity is 340.1 ±2 µS/cm (Meza-Figueroa et al., 2009). Texture analysis revealed that 80% of the material is coarse grained with significant variation in particle size from coarse sand to fine silt (De la O-Villanueva et al., 2013). The most common metals in the center deposit are Fe (31,739 ± 381.9 mg/kg), Cu (400.5 ± 15.8 mg/kg), Rb (298.4 ± 5.6) and Mn (158.5 ± 10.5 mg/kg); mean values of As and Pb are 29.3 ± 4 and 39 ± 4.2 mg/kg respectively (Meza-Figueroa et al., 2009). However, some metals including Cu, Mn, Zn and Ba, reach very high values in efflorescent salts (e.g., Cu: 68,751 ± 865 mg/kg), and are highly susceptible to aeolian transport into nearby residential soils (Meza-Figueroa et al., 2009).

This study focused on the central deposit which has an area of approximately 19 ha, a volume of 1.5 million m3 and a mass of 3.3 million tons (De la O-Villanueva et al., 2013). This tailing is located at 30°22′2.4″N and 109°41′38″W at an elevation of 1,050 masl. The entire deposit was explored for perennial plants resulting in the identification of just four patches that contained all plants. Patch size varied from 34 to 743 m2 for a total of 1,591 m2 or 0.84% of the total area of the deposit.

Plant inventory

The tailings were visited six times during the year in order to collect specimens of perennial species from each of the four patches during the flowering season. Annual species were collected during the summer rainy season. Species were identified using local floras (i.e., for trees: Felger, Johnson & Wilson, 2001) or by comparison with specimens deposited at the University of Sonora Herbarium. Collection was made under Scientific License FLOR-0090 by SEMARNAT.

Abundance and population structure

The abundance of perennial species was determined using 10 × 10 m sampling plots throughout all four patches. Given that patches were of different size, we used as many contiguous plots as necessary in order to cover the entire area of each patch and obtain a census of all perennial plants growing in the tailings. Within each plot the identity of each individual was recorded and its height was measured with a metric tape for shrubs and a graduated telescoping pole for trees. For annual species, a 1 m2 sampling plot was used randomly distributed within each patch. We used a total of six (1 m2) plots per patch. For this set of plants only the identity of each individual and the number of plants per plot were recorded.

Relation between plant distribution and physicochemical properties of tailings

We compared several physicochemical properties of tailings in patches with vegetation and in areas without vegetation. We used a paired sampling approach taking tailings samples (0–20 cm in depth) from each patch with perennial plants and adjacent areas with no vegetation. For each pair we took three randomly located samples that were combined to form a composite sample (1 kg); in total we obtained four composite samples from plant patches and four composite samples from areas without vegetation (n = 8). Once in the lab, samples were dried and homogeneized. The pH and electrical conductivity (EC) were determined in solid suspensions (1:20 solid:water) using a 100 Ecosense pH meter and a portable conductivity meter (Hanna), respectively. Particle size distribution was determined through wet sieving following the method of Beare & Bruce (1993). For metal analysis, homogenized tailing samples were placed in plastic bags and measured directly with a field portable X-ray fluorescence (PXRF) analyzer NITON XL3t. The procedure followed the manufacturer instructions and the recommendations of the EPA-6200 method (US-EPA, 2007). Each sample was analyzed for As, Be, Cd, Co, Cr, Cu, Fe, Mn, Mo, Ni, Pb, Sb, Se, Ti, Tl, V and Zn. We used Till 4 and Montana 2710 as standard reference materials for accuracy and performance checks of PXRF analysis. Values for elements that fell within the ±20% values of the standard were taken as accurate.

Patterns of metal accumulation in plants

The perennial species abundance survey was used to select the five most abundant species for analysis of metal accumulation. Tailing samples (200 g) were collected from the rhizosphere and 5–10 leaves were randomly collected from six individuals of each species distributed in at least two patches. Leaf tissue was rinsed with distilled water several times in the field and transported to the lab. Once in the lab, leaf tissue was rinsed with deionized water and dried at 50 °C for 48 h. Dried leaf tissues were ground in an agate mortar for analysis of metals. Metals in tailing material from the rhizosphere and from leaves were analyzed as previously described (PXRF).

Experiment on local adaptation in an annual and a perennial species

In order to test whether a particular plant species (either annual or perennial) had physiologically adapted to the local tailings conditions, we set up a reciprocal short-term experiment in a shade house at the Instituto de Ecologia UNAM in Hermosillo, Sonora. First, seeds of Prosopis velutina (perennial) and Amaranthus watsonii (annual) were collected from individual plants (four P. velutina trees and ten A. watsonii plants) growing within the tailings deposit and growing in an off-site area close (<100 m) to the deposit. Additionally, tailings and off-site soil samples (5 kg) were taken from each of the sites where seeds were collected. These samples (tailings and off-site soil) were used to fill plastic tubes (5 cm in diameter by 20 cm long); half of the tubes were filled with tailings and the other half with off-site soil. Each family (seeds derived from a single plant) from each site was grown in both tailings and off-site soil. Tubes received either five A. watsonii seeds or one P. velutina seed. Seven tubes per mother (family) for a total of 140 tubes were used for A. watsonii, whereas for P. velutina, from seven to 40 tubes per mother for a total of 226 tubes were used. Tubes were regularly irrigated and the experiment lasted one month. The following parameters were recorded: emergence, height, number of leaves and total, root, shoot and leaf dry mass after 30 d after germination.

Statistical analysis

Plant abundance is expressed as boxplots describing variation of the number of perennial and annual individuals recorded across plots among the four patches. Histograms with the frequencies of different size (height) classes found for each species were used to describe the population structure and the pattern of regeneration of the most abundant perennial species (Silvertown & Charlesworth, 2001). Measured parameters including pH, EC, texture (sand frequency), and metal concentrations in patches with vegetation and areas without vegetation were compared using paired t-tests. We employed the Bonferroni correction for multiple testing using R (R Core Team, 2013). Ratios of metal accumulation were calculated from the mean values found for individuals (leaves/rhizosphere) of all species. A χ2 test was used to determine whether the ratios were significantly greater than 1. For local adaptation, a two-way ANOVA was used to evaluate whether there were significant differences in height, number of leaves and total dry mass due to origin of plants (tailing vs. soil), growth media (tailing vs. soil) and their interaction. We used Tukey HSD as post hoc analysis to evaluate differences between treatments. Local adaptation is inferred if the results show that plants collected from tailings grew better in tailings than in off-site soil and if plants collected from off-site grew better in off-site soil than in tailings. All statistical analyses, except the Bonferroni correction, were performed with JMP (version 3.1; SAS Institute, Cary, North Carolina, USA).

Results

Plant inventory

We recorded a total of 42 species of plants distributed in 16 families of angiosperms (Appendix S1). The most common families were Poaceae and Asteraceae.

Abundance and population structure

A total of 872 individuals of perennial species were recorded growing in four patches in the central tailing. The five most common perennial species were, in order of decreasing abundance: Acacia farnesiana (N = 540), Brickellia coulteri (N = 126), Gnaphalium leucocephalum (N = 108), Baccharis sarothroides (N = 81) and Prosopis velutina (N = 17). The abundance (number of individuals/100 m2) varied among patches with mean values between 0 and 13 individuals/100 m2 (Fig. 1). Among the perennial species, three patterns of population structure were observed: (a) species that are actively regenerating with a large number of the small size classes (seedlings) like A. farnesiana (Fig. 2); (b) species with large number of intermediate size classes (juvenile and adults), like B. coulteri, B. sarothroides and G. leucocephalum that are still regenerating (Fig. 2) and c) species with no evidence of recent regeneration like P. velutina and composed mainly of large size (adults) classes (Fig. 2).

Figure 1 Box plot of abundance of perennial (A–D) and annual (E–H) species recorded at four patches in the central Nacozari tailings deposit.

Boxes indicate the 25–75th percentile, the line within the box shows the median, error bars indicate the 90th percentile and dots indicate outlying points. 1A: patch 1; 1B: patch 2; 1C: patch 3; 1D: patch 4; 1E: patch 1; 1F: patch 2; 1G: patch 3; 1H: patch 4. Perennial species; Acfa: Acacia farnesiana, Gnle: Gnaphalium leucocephalum, Brco: Brickellia coulteri, Basa: Baccharis sarothroides and Prve: Prosopis velutina. Annual species; Amwa: Amaranthus watsonii, Boco: Boerhavia coulteri, Solu: Solanum lumholtzianum and Brca: Bromus catharticus. Notice that unit area is 100 m2 for perennial species and 1 m2 for annual species.

Figure 2 Population structure of the most common perennial species recorded at the center Nacozari tailings deposit.

2A: Acacia farnesiana (n = 540 individuals), 2B: Brickellia coulteri (n = 126 individuals), 2C: Baccharis sarothroides (n = 81 individuals), 2D: Gnaphalium leucocephalum (n = 108 individuals), and 2E: Prosopis velutina (n = 17 individuals).

The most common annual species were Amaranthus watsonii, Boerhavia coulteri, Solanum holtzianum and Bromus catharticus (Fig. 1). The abundance varied among patches with mean values between 0 and 16 individuals/m2.

Relation between plant distribution and physicochemical properties of tailings

There were no significant differences for most measured physicochemical properties between samples taken from patches with vegetation and adjacent areas with no vegetation (pH, EC, texture, and metal content) (Table 1). The correction for multiple testing revealed the same result: there were no significant differences between samples from vegetation patches and barren patches (Table 1).

Table 1 Physicochemical properties (mean ± standard deviation) of the central Nacozari tailing from patches with vegetation and adjacent areas with no vegetation.

Property	Patch with  vegetation	Area with  no vegetation	Statistical test	Significance	Adjusted   p-values*	
pH	4.7 ± 0.2	4.5 ± 0.3	t = 0.97	p = 0.18	0.06	
Electrical conductivity (µS/cm)	162.3 ± 80.1	112.9 ± 10.3	t = 1.06	p = 0.19	0.34	
Percentage of sand	77.8 ± 4.5	79.8 ± 3.7	t = 0.57	p = 0.30	0.66	
Percentage of clay	5.7 ± 2.9	6.1 ± 2.0	t = 0.19	p = 0.57	0.89	
As (mg kg−1)	18.7 ± 2.3	21.0 ± 1.9	t = 0.88	p = 0.79	0.37	
Ba (mg kg−1)	1,091.5 ± 46.8	1,172.5 ± 76.0	t = 1.81	p = 0.06	0.22	
Ca (mg kg−1)	2,872.7 ± 1,779.1	1,286.0 ± 926.8	t = 2.07	p = 0.06	0.10	
Cu (mg kg−1)	333.0 ± 96.3	271.5 ± 45.7	t = 0.05	p = 0.47	0.94	
Fe (mg kg−1)	26,167.7 ± 6,454.6	31,604.2 ± 7,273.2	t = 1.32	p = 0.88	0.10	
K (mg kg−1)	35,741.2 ± 1,247.6	36,821.0 ± 1,557.6	t = 2.25	p = 0.03	0.14	
Mn (mg kg−1)	224.0 ± 22.7	225.7 ± 11.9	t = 0.81	p = 0.77	0.43	
Mo (mg kg−1)	57.2 ± 15.8	65.5 ± 9.1	t = 1.54	p = 0.90	0.22	
Pb (mg kg−1)	32.7 ± 6.5	34.0 ± 8.7	t = 0.26	p = 0.59	0.81	
Rb (mg kg−1)	259.0 ± 9.1	255.0 ± 7.0	t = 0.43	p = 0.34	0.75	
Sr (mg kg−1)	94.2 ± 22.5	84.5 ± 21.4	t = 1.85	p = 0.07	0.28	
Ti (mg kg−1)	1,536.0 ± 208.7	1,394.5 ± 176.4	t = 1.26	p = 0.12	0.20	
Zn (mg kg−1)	69.0 ± 3.4	68.7 ± 1.9	t = 0.64	p = 0.27	0.64	
Zr (mg kg−1)	142.7 ± 4.8	136.2 ± 3.6	t = 1.83	p = 0.06	0.28	
Notes.

* Multiple testing correction using Bonferroni (R Core Team, 2013).

Patterns of metal accumulation in plants

The accumulation factor (leaf/rhizosphere) for most metals in the five most abundant plant species was below 1 (Table 2). The exceptions were for Zn in B. sarothroides and G. leucocephalum where ratios of the accumulation factors were significantly greater than 1. Similarly, for most metals and plant species, leaf metal concentrations did not exceed domestic animal toxicity limits (Appendix S2). For Cu, all plant species reached the domestic animal toxicity limit (15 mg/kg for sheep and 40 mg/kg for cattle; National Research Council, 2005). Similarly, for Mo all plant species also reached the toxicity limit (5 mg/kg for cattle). In addition, for some species and metals, values approached the upper limit of the maximum tolerable range as Zn for G. leucocephalum and Ca for P. velutina (Appendix S2).

Table 2 Metal accumulation factors (mean ± standard deviation) of the most abundant perennial species in the center tailing of Nacozari.

Accumulation factors are calculated as element concentration in leaves/element concentration in the rhizosphere.

Plant species	Baccharis sarotroides	Gnaphalium leucocephalum	Brickellia coulteri	Acacia farnesiana	Prosopis velutina	
Element						
Cu	0.26 ± 0.10	0.8 ± 0.31	0.22 ± 0.08	0.44 ± 0.15	0.51 ± 0.24	
Fe	0.006 ± 0.002	0.13 ± 0.06	0.01 ± 0.005	0.04 ± 0.15	0.05 ± 0.02	
K	1.32 ± 0.43	2.35 ± 0.56	1.57 ± 0.30	0.68 ± 0.16	0.63 ± 0.32	
Mn	0	3.44 ± 2.40	1.25 ± 0.65	0	0	
Mo	0.23 ± 0.06	0.27 ± 0.08	0.19 ± 0.05	0.26 ± 0.06	0.22 ± 0.03	
Rb	0.24 ± 0.08	0.38 ± 0.23	0.21 ± 0.07	0.21 ± 0.03	0.17 ± 0.04	
Sr	0.58 ± 0.62	0.52 ± 0.33	0.51 ± 0.13	0.97 ± 1.0	2.20 ± 1.9	
Zn	3.45 ± 3.6*	9.11 ± 8.5*	1.7 ± 2.6	1.6 ± 1.9	1.55 ± 2.8	
Zr	0.06 ± 0.03	0.11 ± 0.03	0.05 ± 0.006	0.07 ± 0.005	0.064 ± 0.008	
Notes.

Mean ratios (leaves/rhizosphere) for all species were evaluated using a χ2 test.

* Ratios that were significantly greater than 1.

Experiment on local adaptation in an annual and a perennial species

For A. watsonii, seedling emergence varied between 30% and 44% among treatments. After emergence, seedling survival varied between 83% and 90% among treatments. Seedlings growing in tailing material reached 2.8 ± 0.4 to 3.1 ± 0.5 (mean ± standard deviation) cm in height after 30 d whereas seedlings growing in off-site soil reached 7.4 ± 1.3 to 8.2 ± 0.9 cm (Fig. 3A). The statistical analysis revealed a significant difference due to growth medium (F = 426.64, p < 0.001, df = 1∕208), no significant difference due to seed origin (F = 0.71, p = 0.398, df = 1∕208) and a significant interaction (F = 7.39, p = 0.007, df = 1). However, the Tukey test revealed significant differences in height only between substrates and not between sites of seed origin (Fig. 3A). The mean number of leaves from seedlings growing in tailing material was two, whereas seedlings growing in soil had seven leaves (Fig. 3B). As for growth, the statistical analysis indicates a significant difference due to growth medium (F = 391.41, p < 0.001, df = 1∕208), no significant difference due to seed origin (F = 0.01, p = 0.916, df = 1∕208) and no significant interaction (F = 0.52, p = 0.469, df = 1). The Tukey test revealed significant differences in number of leaves only between substrates and not between sites of seed origin (Fig. 3B). For total dry mass, seedlings growing in tailing material accumulated 0.005 ± 0.003 to 0.007 ± 0.004 g and seedlings growing in soil accumulated 0.06 ± 0.01 to 0.08 ± 0.02 g (Fig. 3C). In this case, the analysis shows significant differences due to the growth medium (F = 250.54, p < 0.001, df = 1∕208), seed origin (F = 8.01, p = 0.005, df = 1∕208) and the interaction was significant (F = 11.96, p = 0.001, df = 1). In this case, the Tukey test revealed that seedlings derived from off-site soil and grown in soil accumulated greater biomass than seedlings derived from tailings growing in soil (Fig. 3C).

Figure 3 Height (D), number of leaves (E) and total dry mass (F) of seedlings of Amaranthus watsonii (A, B, C) and Prosopis velutina (F) coming from two different sources (tailings and an adjacent site with normal soil) after 30 days of growth in tailing material and soil.

Different letters indicate significant differences.

For P. velutina, seedling emergence ranged from 67% to 98% among treatments. After emergence, seedling survival varied between 57% and 88% among treatments. After 30 d growth, seedlings growing in tailing material reached 4.5 ± 0.3 to 4.6 ± 0.5 cm whereas seedlings growing in off-site soil reached 8.8 ± 0.3 to 9 ± 0.6 cm (Fig. 3D). The statistical analysis indicates a significant difference due to growth medium (F = 475.38, p < 0.001, df = 1∕317), no significant difference due to seed origin (F = 0.10, p = 0.749, df = 1∕317) and no significant interaction (F = 0.32, p = 0.569, df = 1). The Tukey test revealed significant differences in height only between substrates and not between sites of seed origin (Fig. 3D). For the number of leaves, seedlings growing in tailing material had 3.1 ± 0.9 to 3.7 ± 0.7 leaves whereas seedlings growing in off-site soil had 4.6 ± 1.7 to 7.5 ± 1.6 (Fig. 3E). In this case, a significant difference was detected due to growth medium (F = 91.86, p < 0.001, df = 1∕317), seed origin ( F = 33.84, p < 0.001, df = 1∕317) and a significant interaction (F = 25.19, p < 0.001, df = 1). In this case, the Tukey test revealed significant greater number of leaves in seedlings derived from off-site soil growing in soil than seedlings derived from tailings growing in soil (Fig. 3E). For total dry mass, seedlings growing in tailings accumulated 0.019 ± 0.009 to 0.02 ± 0.003 g whereas seedlings growing in off-site soil accumulated 0.06 ± 0.01 to 0.07 ± 0.02 g (Fig. 3F). As for number of leaves, a significant difference in dry mass was detected due to growth medium (F = 192.46, p < 0.001, df = 1∕317), due to seed origin (F = 15.15, p < 0.001, df = 1∕317) and a significant interaction (F = 10.51, p = 0.001, df = 1). The Tukey test revealed significant greater biomass in seedlings derived from off-site soil growing in soil than seedlings derived from tailings growing in soil (Fig. 3F).

Discussion

Successful implementation of phytostabilization on mine tailings, the establishment of a permanent vegetative cover, requires careful consideration of the plants to be used. This ideally includes a combination of different perennial species with different rooting depths and canopy cover structure that do not accumulate metals into shoot tissues (Mendez & Maier, 2008). In addition, the geographic range of plant species must be taken in consideration, making plants that can naturally colonize mine tailings of interest for their phytostabilization potential. This study identified 42 different species of plants growing in small patches on the abandoned Nacozari mine tailings. Fifteen of these species were perennial whereas twenty seven were annual. From the set of fifteen perennial species, only five were abundant and from this set, four of the five showed clear evidence of regeneration, i.e., the ability to reproduce and establish new seedlings. This set of perennial species normally produces seeds almost every year (F Molina-Freaner, pers. obs., 2016) but recent seedling establishment is restricted to A. farnesiana, B. coulteri, B. sarothroides and G. leucocephalum. We do not know the mechanism that restricts recent seedling establishment in P. velutina and future studies should identify the regeneration barriers of perennial species in this tailings. Taken together, these results suggest that this set of species has potential for phytostabilization of the Nacozari tailings, as they include trees (P. velutina and A. farnesiana) with relatively large canopy cover and deep roots, and shrubs (B. coulteri, B. sarothroides and G. leucocephalum) with smaller cover and relatively shallower roots (F Molina-Freaner, pers. obs, 2016).

Annual species identified on this site had greater diversity (27/42) than the perennial species (15/42). However, annual species have a more limited potential for phytostabilization given their shallow roots, short life cycles and the fact that they grow only during the summer rainy season in this region (Shreve & Wiggins, 1964). However, they may regularly add organic matter to the tailings facilitating the establishment of perennial species and reducing erosion.

Identifying factors that limit plant establishment in tailings is an important goal in phytostabilization (Parraga-Aguado et al., 2013). This study recorded only four patches of vegetation on the tailings, representing 0.84% of the 19 ha surface area. These patches which contained 872 individuals of perennial species have very slowly colonized the tailings over more than 6 decades since the mine was closed in 1949. This suggests there are key inhibitory factors preventing seed germination and seedling establishment at this site. Paired sampling comparing patches of vegetation with adjacent barren areas revealed no differences in pH, electrical conductivity, texture or the concentration of potentially toxic elements. So, these factors do not influence the distribution of plants in the four patches at this site. Future studies should explore whether organic matter, major nutrients (such as N or P), metal bioavailability, moisture content, or the composition of microbial communities restrict plant establishment in this site to these four small patches. A preliminary study suggests major differences in organic matter and N between the four vegetation patches and the adjacent barren areas (L Arvizu and F Molina-Freaner, 2016, unpublished data).

Plants used in phytostabilization should be metal tolerant but limit metal accumulation into above ground tissues in order to avoid transferring toxic elements into food chains (Mendez & Maier, 2008). The five most abundant plant species from the Nacozari tailings had metal accumulation ratios that were generally below 1. The exception was Zn for which two perennial species had accumulation factors greater than 1. Thus, of the five most abundant perennial species, three comply with the requirement of low accumulation. In addition, plants suitable for phytostabilization should have metal concentration in their leaves below the maximum tolerable level for animals (Mendez & Maier, 2008). In our study, we recorded that the five most abundant perennial species have Cu and Mo concentration in their leaves above animal toxicity levels (National Research Council, 2005). Given the accumulation of Cu, Mo and Zn in the most abundant species, future studies should explore whether amendments such as compost, could reduce metal accumulation and meet requirements for animal toxicity (Solis-Dominguez et al., 2012). It is important to note that plant metal uptake into above ground shoot tissue is not necessarily related to the total amount of metals in tailings; rather, it is related to the bioavailable metal concentration in the tailings (Perlatti et al., 2015) and also to the efficiency of the metal transport system in each plant (Verkleij et al., 2009). Thus, both mine tailings and plant species characteristics ultimately influence plant metal uptake and will have to be evaluated on a site-specific basis when using phytostabilization technology.

The success of phytostabilization depends on the use of locally adapted plants. It has been previously suggested that using locally adapted ecotypes provides better cover and persistence than other sources of seeds when grown in mine wastes (Smith & Bradshaw, 1979). Our short-term experiments testing local adaptation in an annual and a perennial species did not detect strong evidence of tailings ecotypes even after a period of more than 60 years. Our data reveal that tailings as a growth substrate significantly reduced the amount of biomass produced when compared with plants grown in off-site soil. The effect of seed origin was contingent upon substrate, as seedling derived from plants growing in off-site soil had equal or greater performance than those derived from tailings plants when grown in soil. However, there were no significant differences due to seed origin when grown in tailings substrate. We suggest that differences in the maternal environment associated with abiotic stress (metals in tailings) might have an effect on progeny performance in seedlings coming from tailings. Similar patterns have been observed in other species (Sultan, Barton & Wilczek, 2009; Zas, Cendan & Sampedro, 2013). However, both the annual and perennial species tested show that seeds harvested from plants growing on the tailings site or in a neighboring off-site area are equally effective for phytostabilization in the Nacozari tailings. Thus, we suggest that future field studies in the Nacozari tailings should use regionally-adapted sources of seeds.

In conclusion, this plant inventory identified several native perennial species with potential for phytostabilization of the Nacozari mine tailings. One advantage of using these plants is that seed could be collected locally and used to revegetate the site. However, two of the species identified accumulate Zn and all species accumulate Cu and Mo into the range of maximum tolerable levels for animals (National Research Council, 2005). Thus, measures should be taken to evaluate and minimize the risk of transferring toxic elements into the local food chain.

Supplemental Information

Data S1 Raw data on the plants from the Nacozari tailings

Raw data on the abundance of perennial and annual plant species, physicochemical properties of tailings, growth of Amaranthus watsonii and Prosopis velutina and metal concentration in rhizosphere and leaves of plants growing in the Nacozari tailings.

Click here for additional data file.

Appendix S1 Click here for additional data file.

Appendix S2 Click here for additional data file.

We thank Jose Martinez for field and lab assistance.

Additional Information and Declarations

Competing Interests

Author Contributions

Field Study Permissions

Data Availability

The authors declare there are no competing interests.

Alina E. Santos conceived and designed the experiments, performed the experiments, analyzed the data, wrote the paper, prepared figures and/or tables.

Rocio Cruz-Ortega conceived and designed the experiments, wrote the paper, reviewed drafts of the paper.

Diana Meza-Figueroa conceived and designed the experiments, reviewed drafts of the paper.

Francisco M. Romero analyzed the data, contributed reagents/materials/analysis tools, reviewed drafts of the paper.

Jose Jesus Sanchez-Escalante analyzed the data, reviewed drafts of the paper.

Raina M. Maier contributed reagents/materials/analysis tools, wrote the paper, reviewed drafts of the paper.

Julia W. Neilson contributed reagents/materials/analysis tools, reviewed drafts of the paper.

Luis David Alcaraz wrote the paper, reviewed drafts of the paper.

Francisco E. Molina Freaner conceived and designed the experiments, performed the experiments, analyzed the data, contributed reagents/materials/analysis tools, wrote the paper, prepared figures and/or tables, reviewed drafts of the paper.

The following information was supplied relating to field study approvals (i.e., approving body and any reference numbers):

The permit (Flor 0090) was received from the Secretaria del Medio Ambiente y Recursos Naturales.

The following information was supplied regarding data availability:

The raw data has been supplied as a Supplementary File.

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
