# Peer review of "Plants from the abandoned Nacozari mine tailings: evaluation of their phytostabilization potential"

_PeerJ, doi:10.7717/peerj.3280_

## Round 0.1 · original submission · Major Revisions

First of all apologies for the delay in getting this decision back to you, unfortunately the lack of availability of one reviewer intersected with my own absence.

In general I found this to be a very well-written and thoughtfully executed study that makes an important contribution to knowledge regarding the restoration of mine tailing sites. The two reviewers both agree that the paper is suitable for publication in PeerJ but have made detailed and constructive suggestions for you to consider. I concur with the reviewers' assessments.

In addition to the reviewers comments I noted the following important points as I read the manuscript:

P4 L8 - I would question the importance of nativity. They certainly shouldn't be invasive but in highly modified "novel ecosystems" (sensu Hobbs et al. 2009; see http://dx.doi.org/10.1016/j.tree.2009.05.012) might non-native species be considered if they provide appropriate ecosystem function and desirable levels of phytostabilisation?

p9 L2 - Please describe how the locations for the 10 x 10 m plots were selected. It sounds like there were differences in sampling intensity between patches. That and the differences in patch size will affect the number of plant species present in each plant (cf. species-area relationship). What implications, if any, might this have for your analysis?

I'm also interested in why you chose to census the number of plants rather than looking at total vegetative cover. It might be fair to assume that the two are roughly correlated but it may also depend on what the dominant species is. Might it be interesting to look at commonly-used indicators of ecological function such as total cover, or the cover of different plant functional groups?

P13 L12 - Statistical results should be reported in full (F, p and degrees of freedom - numerator and denominator). A table presenting these results would be preferable. Your analysis should also examine and report the effect of the interaction of seed and soil origin.

P16 L9 - Regarding limits on plant colonization, it could also just be a stochastic process mediated by the need for regeneration microsites and the distance from nearby seed sources. The latter is a major barrier. Would it be interesting to examine i) seed dispersal and distribution across the site (i.e. seed rain and seed bank structure); and ii) processes of competition and facilitation that might control vegetation establishment and community assembly following initial colonization?

P17 L14 - Regarding use of local seed (what I interpreted to mean seed from plants within the tailings), with no obvious benefit of using seeds from plants on the tailings I would suggest that ensuring genetic diversity at the site by using regionally-adapted seed will be more important. Only using seed from the site itself could lead a genetic bottleneck and reduce overall fitness and resilience in the future.

Figure 1 - I would prefer to see this data presented as boxplots. At the least you should indicate what the error bars are (SE or SD) and show them above and below the mean

Figures 3 and 4 could be combined

Table 1 should be moved to an Appendix

My slight concern with Tables 2 and 3 is that you've completed a large number of individual statistical tests here. A number of these are likely to yield significant results purely by chance. I wonder if a multivariate approach to examining these differences would be more robust? Alternatively you might consider applying a Bonferonni correction (though that can be pretty conservative). Might an alternative be to model metal concentrations using a linear model that includes element and vegetation status as interacting factors?

Please also refer to the attached annotated pdf on which I've noted a number of more minor suggestions, queries and corrections.

Reviewer 1 ·

Basic reporting

In general, the manuscript is well focused, supported in literature well referenced and relevant, and coherent among sections. There is, however, a relevant aspect missing in the context and discussion of the manuscript. This is related to the fact that not all mine tailings pose metal toxicity problems to plants as metal bioavailability broadly varies on these substrates. In the case of studied tailings dump it seems to be high bioavailability of metals as secondary acidification occurred (very low pH measured), but this is not the case in all tailings dumps. There is a site-specific condition in terms of metal bioavailability that needs to be incorporated in introduction and discussion. Metal tolerant plants will be only developed (evolve) on sites where metals are bioavailable. This also needs to be considered to really provide a “methodology that can be used to identify native plants and evaluate their phytostabilization potential for any mine tailings site” as stated by authors at the end of the abstract. Measurement of metal translocation to aerial parts of the plant cannot be used as an indicator of metal bioavailability in tailings or in metal polluted soils. It is only an adequate indicator for choosing proper plant species for in situ stabilization of the site and protection of herbivores that might be present in the area.

The manuscript uses professional English throughout and its structure conforms to PeerJ standards. Raw data is supplied (see Appendix 1) and all figures and tables are relevant and well labeled and described, in general with the following minor request:
- Figure 1: Add at the end of the legend “Mean values and standard deviation are given”.
- Figure 2: size of texts in both axes should be increased.
- Figures 3 and 4: they can be fused into one figure (as figure 1), with graphs of one species in one column and the other species in a second column. Furthermore, size of text in axe y should be increased and species name on top should be deleted and indicated in the legend.
- Table 1: species names have to be in italics
- Appendix 1: Add at the end of the legend “Mean values and standard deviation are given”.

Experimental design

Research question is well defined as so is the knowledge gap. It is relevant and meaningful. However, site-specificity of the addresses problem should be strongly stated and considered. The case study site should be acknowledged as an example but there is a broad range of conditions that vary from site to site, not clearly stated in the manuscript. Therefore, conclusions should be carefully stated in terms of site specific conditions/characteristics of study site and more general ones.
Methods and experimental design are well described with sufficient detail and information to replicate. It is a rigorous investigation performed to a high technical and ethical standard.
However, even though statistical analyses are clearly stated, there is a lack of statistical results on data related to abundance of plants among and inside patches (figure 1). It should be included. In the case of the experiment on local adaptation in an annual and a perennial species, a 2-way ANOVA was performed but no indication of interaction factors is given. Indicate.
In the case of the plant inventory, consideration of the origin (native, endemic, exotic) of the species may be included as it is very relevant information for the goal of the study.

Validity of the findings

In general, findings described in the manuscript are valid; data is robust, statistically sound and controlled. However, some considerations should be incorporated at discussion. For example, are there natural and/or introduced herbivores in the area which may be exposed to bioaccumulation problems? There is no reference support for the root systems of the species evaluated or with potential (see lines 12 to 16 on page 15). Indication of “annual species have a more limited potential for phytostabilization given their shallow roots, short life cycles and the fact that they grow only during the summer rainy season in this region” (lines 19 to 21 page 15) misses the fact that standing dead biomass of annual plants also allows erosion control. Reconsider this statement. Discussion is mainly focused on potential limiting chemical and physical characteristics of tailings but it missed aspects related to microsite conditions or safe sited for regeneration, such as cracks. Please, consider these other microsite conditions too. Finally, as drought tolerant plants present in the study site may have morphological structures (trichomes, resin glands, etc.) to protect them in this condition, even though washing before analytical determinations were performed in leaves, external pollution may still be present in them. This should be mentioned in the text. Finally, as indicated above, consideration of the metal bioavailability concept has to be incorporated at discussion section.
Conclusions are well stated but site-specific conditions should be considered. How this local study could help in a general and international context have to be stressed and better considered.

Additional comments

No extra comments

Reviewer 2 ·

Basic reporting

The manuscript is well written. Tables and Figures adhere to policy

Experimental design

sound experimental design.

Validity of the findings

The findings are supported by data presented. The authors could have gone further. I have detailed my comments below to the author

Additional comments

This is a novel contribution for phytostabilization of mine tailings in a arid/semi-arid environment. A few additions would help the reader to get more from the findings. I have detailed these below.

are the plants investigated halophytes? Soil EC of 16.3 dS/m show this is a very saline soil. How are these plants tolerating this salinity? Are they salt tolerant? If not, can you expect sustained growth of candidate plant species?

Are the plant species acid or metal tolerant? Soil pH <5 can present phytotoxicity problems in metal contaminated soil. Are the plant species tolerant to soil solution Cu which would be expected to be elevated and phytotoxic in a Cu tailing at pH < 5.

Soil metal levels are reported for contaminated (with/without vegetation) areas but not surrounding natural background soils. A comparison of the contaminated soil with natural metal background soils would provide valuable information on the long term potentail success of the phytostabilization. For example, are local native plant species or invasive plants excluded because of high metal phytoavailability? or salinity? You can obtain background soil metal levels from USGS for Arizona at http://pubs.usgs.gov/ds/801/ .

Rubidium is a rare odd element to have in elevated amounts unless it is specific to the mine ore tailings. You may want to check your XRF Rb data again.

---

## Round 0.2 · Minor Revisions

Thanks for the changes you have made to your manuscript and for your very thorough response to the reviewers' comments. I agree with the reviewers that the paper has been much improved as a result of your hard work. There are, however, a few remaining issues that I would request you give your attention to:

FIGURES
1. The figures provided do not seem to display properly when downloaded and appear to have a significant proportion of the image cropped out. Please check them carefully
2. Ensure fonts are consistent across figures (e.g. F1 and 2 are different)
3. In the revised version of Figure 3 the shading of the columns needs to be changed to that both the positive and negative error bars are visible. A box plot would probably be preferable in this situation too.

SITE DESCRIPTION
3. In their initial comments, Reviewer 2 requested information from surrounding soils be included for comparative purposes. I realize you only have 6 measurements but please include a summary of this information in the paper (for instance in the site description section of the Methods)
4. Please provide a Lat/Long for the study site location
5. With regards to providing a description of the vegetation in the local area (i.e. the reference conditions for any restoration efforts), simple identification of local dominant species would be helpful to provide ecological context. I'd be surprised if there weren't local botanical or ecological surveys which could provide information on general plant community types. The following look like they might be of use:
https://www.researchgate.net/profile/Alberto_Burquez/publication/239522466_Vegetation_and_habitat_diversity_at_the_southern_desert_edge_of_the_Sonoran_Desert/links/0deec5355a7b9001ab000000.pdf
http://wildsonora.com/sites/default/files/reports/the-vegetation-and-flora-of-the-region-of-the-rio-de-bavispe-stephen-white-1948.pdf
https://www.jstor.org/stable/43781269?seq=1#page_scan_tab_contents

METHODOLOGY
6. A nice description of the contiguous plot monitoring method is provided in the response, I would like to see this information provided in the Methodology to ensure clarity over the sampling design.
7. Note importance of careful interpretation when statistical results reveal a significant interaction. One can't say a factor has an effect (or not). Rather the effect is contingent upon the level of the interacting factor. Appropriate post-hoc testing can help unravel what's happening. See for example: http://pages.uoregon.edu/stevensj/interaction.pdf

RESULTS
8. Please correct variation in number of decimal places in the reporting of statistical results (e.g. p14 - p-values vary from 1-4 places. Small p-values can be reported as p < 0.001. Reporting should be consistent).
9. With so many statistical tests, and a desire to easily compare effects on different plant traits, I would like to see these statistical results placed in a table. Should p-values be corrected to account
for the large number of tests?

Reviewer 2 ·

Basic reporting

fine

Experimental design

fine

Validity of the findings

good

Additional comments

The authors have addressed all my comments in a highly satisfactory manner

---

## Round 0.3 · Minor Revisions

Many thanks for your efforts to address the comments from the previous review. In general I think your paper is very close to being ready. I just had a couple of queries regarding some of the revisions you've made:

1) Thanks for including correct p-values for the t-tests to account for the issues of multiple testing. I'm not familiar with the method proposed but a brief review of the reference cited (White et al. 2009) suggests it was developed for count (frequency) data. Can you describe how the method is appropriate for your soil chemistry data - there seem to be some surprisingly large shifts in a number of the p-values following corrections. Why not use the more familiar Bonferonni correction?

2) In the vegetation description note that species names should be given in full at first mention even if the genus has previously been written in full in the context of a different species. For example (P7 Line 11) "Acacia constricta, A. farnesiana, Fouquieria splendens..." should be "Acacia constricta, Acacia farnesiana, Fouquieria splendens..."

3) For the ANOVA tests degrees of freedom are missing for some of the stated results. Again consider the importance of interactions - for instance the effect of seed origin is contingent upon the substrate type. What's interesting is that seedling from soil plants performed the same or better than those from tailings plants when grown in soil but there is generally no difference when grown in tailings. Why might this be? What might limit the growth of tailings-origin seedlings when grown in soil?

---

## Round 0.4 · accepted · Accept

Many thanks in your efforts in dealing with all the comments from the reviewers and myself. I appreciate your patience and the constructive manner in which you engaged with the suggestions. In a final version you may wish to consider inserting a line or two to suggest what might explain the distribution of vegetation on the tailings given that there was no significant difference in any of the soil parameters. Is it to do with random establishment in favorable microsites?